# Entomological Surveillance Activities in Regions in Greece: Data on Mosquito Species Abundance and West Nile Virus Detection in *Culex pipiens* Pools (2019–2020)

**DOI:** 10.3390/tropicalmed8010001

**Published:** 2022-12-20

**Authors:** Annita Vakali, Stavroula Beleri, Nikolaos Tegos, Anastasia Fytrou, Anastasia Mpimpa, Theodoros N. Sergentanis, Danai Pervanidou, Eleni Patsoula

**Affiliations:** 1Hellenic National Public Health Organization, 15123 Athens, Greece; 2Laboratory for Surveillance of Infectious Diseases, Department of Public Health Policy, School of Public Health, University of West Attica, 11521 Athens, Greece

**Keywords:** Greece, West Nile virus, mosquitoes, *Culex pipiens*, entomological surveillance

## Abstract

Human cases of West Nile virus (WNV) infections have been recorded in Greece since 2010, with seasonal outbreaks (summer-autumn) on an almost annual basis, caused mainly by the WNV lineage 2 strain (Nea Santa-Greece-2010). National Public Health Organization (NPHO) in Greece is annually implementing enhanced surveillance of human WNV infection, in order to promptly identify human cases of WNV infection and monitor distribution in terms of time and place. Entomological surveillance activities were carried out on a national basis in 2019 and 2020, under NPHO coordination and the collaboration of several private subcontractors, along with the Unit of Medical Entomology, Laboratory for Surveillance of Infectious Diseases (LSID). The aim was to monitor mosquito species composition, abundance, and WNV circulation in mosquito pools of *Culex pipiens* s.l. species. Adult mosquito traps were placed in selected sites; collected samples were morphologically characterized and pooled by date of collection, location, and species types. Female *Culex pipiens* s.l. pools were tested for WNV and WNV infection rates (MIR and MLE) were estimated. Highest mean number of female *Culex pipiens* s.l. species was recorded in Central Macedonia both for 2019 and 2020. Six hundred and fifty-nine mosquito pools (147 in 2019 and 512 in 2020) of female *Culex pipiens* s.l. were examined for WNV presence. The highest MLE was detected in Western Macedonia in 2019 and in Thessaly in 2020. Here, we present data on the mosquito species composition in the studied areas and WNV detection in mosquitoes from areas in Greece where the specific national mosquito surveillance program was implemented, for two years, 2019 and 2020.

## 1. Introduction

West Nile virus (WNV), family Flaviviridae, genus Flavivirus, is transmitted to humans, equids, and other mammals through the bites of infected mosquitoes, mainly of the *Culex* genus [1]. WNV presents a high biological diversity and up to nine distinct lineages have been proposed so far. Only lineage 1 and 2 West Nile viruses have been associated with significant outbreaks in humans. Birds are the reservoir hosts, while humans and equids are considered dead-end hosts [2].

Mosquitoes (Diptera: Culicidae) are the most abundant arthropod groups with public health importance [3]. *Culex*, *Aedes,* and *Anopheles* genera are of particular medical interest, as they are considered transmission vectors of parasites and viruses associated with a significant number of human and animal diseases. Vector-borne diseases account for more than 17% of all infectious diseases worldwide [4,5]. WNV is maintained in nature in a cycle between mosquitoes and animal hosts, humans, equids, and other mammals, through the bites of infected mosquitoes, mainly of the *Culex* genus. It presents a wide distribution throughout Africa, the Middle East, southern Europe, western Russia, southwestern Asia, and Australia, based on its ability to infect multiple mosquito and bird species [6,7].

Annual seasonal outbreaks during summer and early autumn months along with a number of locally acquired human cases of WNV infections are recorded in each transmission season, including years 2019 and 2020 [8], thereby constituting WNV infection an emerging public health challenge in the European Union (EU).

Human cases of WNV infection have been recorded in Greece since 2010, with seasonal outbreaks occurring almost annually [9,10]. Enhanced surveillance has been implemented ever since, as a necessary public health measure for early detection of human cases and prompt implementation of targeted response actions. The largest number of human cases of WNV infection ever recorded in Greece occurred in 2018, with 317 recorded human cases and 86 affected municipalities, indicative of intense viral circulation [9,10]. These data are in accordance with the 2018 surveillance data at the European level, when the highest-ever number of human cases was recorded [11].

WNV outbreaks have been associated with increased mosquito populations. Although the virus has been isolated from over 40 different mosquito species (Diptera: Culicidae), the principal genus involved in WNV transmission is considered to be *Culex*, particularly the *Culex pipiens* complex [12]. European populations of *Cx. pipiens*, biotypes *pipiens* and *molestus*, are both susceptible to WNV infection and capable of transmission [13]. Based on its vector competence and the frequency of natural infection, *Cx. pipiens* is considered to be a major vector in WNV transmission. *Cx. pipiens* form *pipiens*, which mainly feeds on birds, probably plays an important role in the enzootic cycle, whereas the form *molestus* and hybrids which feed on both birds and mammals are more likely to play the role of a bridge vector in the epizootic cycle [14,15]. Additionally, *Aedes albopictus*, *Aedes detritus,* and *Culex modestus*, have been experimentally confirmed as competent species for transmission of WNV [16,17,18].

In Greece, WNV outbreaks have been associated with *Cx. pipiens* s.l., the most abundant *Culex* species in the country, considered to be involved in the transmission of the virus [19,20,21]. Here, we present data on mosquito species and population abundance and detection of WNV in *Cx. pipiens* s.l. pools, from entomological surveillance activities, in the context of the national mosquito surveillance program coordinated by the National Public Health Organization (NPHO), implemented in several Regions (R, NUTS2 level) and Regional Units (R.U., NUTS3 level) of Greece, during two seasons of mosquito activity, in 2019 and 2020. The aim was to emphasize the importance of recording mosquito species circulating and the early detection of WNV circulation in mosquito vectors at the local level.

## 2. Materials and Methods

### 2.1. Regions, Mosquito Collection and Identification

NPHO organized and funded mosquito surveillance activities in Greece, for the years 2019 and 2020, in regions (R, NUTS2 level), and regional units (RU, NUTS level 3). Two types of traps, CO_2_ mosquito traps with dry ice (CDC Miniature Light Trap; BioQuip Products, Inc., Rancho Dominguez, CA, USA) and Mosquito Triple Traps (Zhongshan Skone Electric Appliance Co., Ltd., Zhongshan, China), suitable for adult mosquito collections, were used.

In 2019 (September–November): Traps were placed in 34 stable sites, median: 34 (range: 6–40) traps/ISO week, in 22 RU and 8 R.

In 2020 (August–November): Traps were placed in 59 stable sites, median: 54 (range: 9–62) traps/ISO week, in 38 RU and 11 R.

Traps were placed by private subcontractors of NPHO, in both urban and rural areas of the R.U.s included in the National Entomological Surveillance Program, in cooperation with NPHO for optimal representation of different ecotypes. All the captured insects from each mosquito trap were collected, placed in dry ice, and shipped to the laboratory of the Medical Entomology Unit, located in the School of Public Health, University of West Attica, Athens, Greece, for morphological identification, counting of mosquitoes and WNV molecular detection. Identification of mosquitoes, based on morphological characters, was performed after detailed examination under an SMZ-U Nikon binocular scope, using appropriate dichotomous keys [2,22,23]. Adults morphologically characterized as *Anopheles maculipennis* and *Anopheles sacharovi* were further examined by molecular amplification methods, according to previously described protocols [24,25]. Twelve regions were included in the National Entomological Surveillance Program of NPHO, for the 2019 and 2020 seasons of mosquito activity (Figure 1A and Figure 2A). Regions of Attica, Central Greece, Central Macedonia, Crete, Thessaly, Western Greece, and Western Macedonia were included in the program in both years (Figure 1A and Figure 2A), the region of Peloponnese only in 2019 (Figure 1A), and the regions of East Macedonia and Thrace, Epirus, Ionian Islands, and North Aegean only in 2020 (Figure 2A).

A total of 987 traps were placed and examined during the two years; in particular, 357 traps were placed in eight Regions in 2019 (Figure 1B), from 14 September 2019 (ISO Week 37) to 28 November 2019 (ISO Week 48), and 630 traps were placed in 11 regions in 2020 (Figure 2B), from 18 August 2020 (ISO Week 34) to 10 November 2020 (ISO Week 46), respectively.

### 2.2. WNV in Mosquito Pools

Mosquitoes morphologically identified to species level as *Cx. pipiens* s.l. females were pooled in a single tube according to date and location, with a minimum of two and a maximum number of 150 adults per pool. Pooled mosquitoes were stored and frozen at −80 °C. Genetic material (RNA) from mosquito pools was extracted using the Maxwell 16 Automated Nucleic Acid extraction system (Promega, Madison, WI, USA), along with the Maxwell 16 LEV Simple RNA Tissue kit, as previously described [20]. For WNV detection, the TaqMan real-time PCR protocol of Tang et al. [26], specific for WNV lineages 1 and 2, was implemented. For further verification of the WNV-positive pools detected with this method, another real-time PCR protocol, targeting the nonstructural NS2A region was also applied [27]. An RNA sample of WNV was used as a positive control marker, whereas negative controls, including RNA samples from several arthropod-borne viruses such as Dengue and Chikungunya viruses, were also included in the reaction (BEI Resources ATCC).

### 2.3. Statistical Analysis

All collected data were entered into an Excel database. Minimum infection rate (MIR) and maximum likelihood estimation (MLE) were calculated using the PooledInfRate, version 4.0, CDC’s Excel add-in [28]. Descriptive statistics (means and standard deviations, denoted as SD) were calculated for numerical variables; absolute and relative frequencies were used to summarize categorical variables. Between-region heterogeneity in the mean numbers of female *Cx. pipiens* s.l. mosquitoes collected per sampling was evaluated using Kruskal–Wallis test; post hoc, each Region was compared versus the Region with the lowest mean using the Mann–Whitney–Wilcoxon test for independent samples. Whenever data were available for both years, the differences in the mean numbers of collected female *Cx. pipiens* s.l. mosquitoes per sampling between 2019 and 2020 were assessed separately for each Region using the Mann–Whitney–Wilcoxon test.

In addition, the between-region heterogeneity and 2020 vs. 2019 comparisons in the Np/Nt rate (i.e., number of WNV positive pools/numbers of tested pools) were evaluated with Fisher’s exact test, as appropriate. The level of statistical significance was set at 0.05. Statistical analysis was performed with STATA/SE version 16.1 (Stata Corp., College Station, TX, USA).

## 3. Results

A total of 56,120 adult mosquitoes were captured in the 12 regions included in the National Entomological Surveillance Program of NPHO, for the 2019 and 2020 seasons of mosquito activity. In 2019, a total of 12,282 (92,9% of which were females) adult mosquitoes were trapped belonging to 6 genera and 14 species, and in 2020 a total of 43,838 (94% of which were females) adult mosquitoes were trapped, belonging to 16 species and 6 genera (Figure 3). Species’ composition of mosquito assemblages was different between regions; however, *Cx. pipiens* s.l. (58.15%) and *Aedes caspius* (32.35%) were by far the most abundant of all the collected species (Figure 3). The largest mean number of total mosquitoes per sampling in 2019 was collected in Central Macedonia, followed by Western Greece while in 2020 the largest median number of mosquitoes was collected in Western Greece, followed by Central Macedonia. A total of 15 mosquito species, recorded in 2019 and/or 2020, are known vectors of pathogens. Four (4) of the detected species, namely *An. sacharovi*, *An. maculipennis*, *Cx pipiens* s.l., and *Ae. albopictus* are of major medical importance.

*Ae. albopictus* were captured in 7 out of 8 regions surveyed in 2019 and in 10 out of 11 in 2020. The largest mean numbers of *Ae. albopictus* mosquitoes per sampling were collected in 2019 in Central Macedonia (1.42) followed by Crete (1.27) and in 2020 in Attica (1.84) followed by Central Macedonia (1.71). 

Regarding *Ae. caspius*, the abundance was higher in Western Greece and Central Macedonia, and it was especially high in Western Greece in 2020 (mean number of 205.42 mosquitoes per sampling), when it represented 84% of the total mosquitoes collected.

In 2019, we did not collect any *An. maculipennis* nor *An. sacharovi*. A small number of *An. claviger, An. hyrcanus* and *An. algeriensis* were collected in six, two, and three regions, respectively (Figure 3).

In 2020, *An. maculipennis* were collected in East Macedonia and Thrace, Thessaly, and Central Macedonia with a mean number per sampling of 1.46, 0.14, and 0.02, respectively. *An. sacharovi* was collected in East Macedonia and Thrace, Thessaly, and Western Greece with a mean number per sampling of 0.17, 0.08, and 0.04, respectively. A small number of *An. claviger*, *An. hyrcanus* and *An. algeriensis* were collected in seven, five, and four regions, respectively (Figure 3).

The largest mean numbers of *Cx. pipiens* s.l. mosquitoes per sampling both in 2019 and 2020 were collected in Central Macedonia (Figure 3 and Table 1). Regarding the abundance of the WNV vector (female *Cx. pipiens* s.l. mosquito), the mean (SD; range) numbers per sampling, by year and region, are presented in Table 1.

Regarding 2019 (Table 1), the between-region heterogeneity in the mean numbers of female *Cx. pipiens* s.l. mosquitoes collected was statistically significant (*p* = 0.0001, Kruskal–Wallis test); the highest mean number was collected in Central Macedonia (mean = 60.19; SD = 153.30), followed by Thessaly (mean = 15.33; SD = 32.14) and Attica (mean = 9.67; SD = 22.04). All regions except for Central Greece differed significantly from the region with the lowest mean (Western Macedonia).

In 2020, the between-region heterogeneity was again statistically significant (Table 1, *p* = 0.0001, Kruskal–Wallis test); the highest mean number of female *Cx. pipiens* s.l. mosquitoes was collected in Central Macedonia (mean = 143.79; SD = 269.19), followed by Thessaly (mean = 58.17; SD = 200.23), and Western Macedonia (mean = 30; SD = 26.46). All regions except for Central Greece and Epirus differed significantly from the region with the lowest mean (North Aegean).

Concerning the 2020 vs. 2019 comparison (Table 1), a significant increase was noted in Western Greece (*p* = 0.0001), Thessaly (*p* = 0.002), and Central Macedonia (*p* = 0.027), whereas a significant decrease was observed in Attica (*p* = 0.0005); no significant changes were documented in Central Greece (*p* = 0.551) and Crete (*p* = 0.265).

During the 2-year program, 29,606 female *Cx. pipiens* s.l. mosquitoes (5971 collected in 2019 and 23,635 collected in 2020), captured, were pooled and tested for WNV (Table 1). Out of 659 mosquito pools tested (147 in 2019 and 512 in 2020), a total of 30 (4.6%) were found positive for WNV, 18 (12%) in 2019 and 12 (2%) in 2020, with both real-time PCR protocols implemented. Regions with positive mosquito pools and total number of WNV-positive pools per region, per sampling year, are shown in Table 2.

A total of 30,808 female *Cx. pipiens* s.l. mosquitoes collected in both years (7021 in 2019 and 23,787 in 2020) were grouped in 659 pools and tested for WNV. Out of 659 tested pools, a total of 30 (4.6%) mosquito pools were found positive for WNV. The number of tested and positive for WNV *Culex pipiens* s.l. mosquito pools, MIR and MLE, per region, per sampling year, is shown in Table 2.

Despite numerical differences in Np/Nt rates, the overall between-region heterogeneity did not reach statistical significance in 2019 (*p* = 0.577, Fisher’s exact test) or in 2020 (*p* = 0.270, Fisher’s exact test). A significant decrease in the Np/Nt rate was noted in Central Macedonia in 2020 vs. 2019 (*p* = 0.002); no significant differences were noted in any other region with available data for both years (Table 2).

In the context of the national entomological surveillance program, positive *Cx. pipiens* s.l. pools for WNV RNA were detected in 2019 in three out of the four regions with human cases, and in 2020 in two out of the four regions with human cases.

## 4. Discussion

Greece is a European country where WNV infection cases are recorded on an almost annual basis since 2010, suggesting that the pathogen is established in the country. Recurrence of WNV infection cases in Europe is expected in each transmission period, during the mosquito circulation season. However, due to its complex epidemiology, WNV’s circulation and geographical distribution cannot be safely predicted, rendering the need for established and sustainable surveillance systems of WNV circulation, including vector surveillance and control activities—at the national, regional, and local level—of major importance.

Under the auspices of an entomological surveillance program coordinated by NPHO, a program targeting monitoring of mosquito abundance, species composition, and the circulation of WNV in mosquito pools of the species *Cx. pipiens* s.l. operated on a national basis for the years 2019 and 2020. The geographical coverage of the surveillance was wider than the previous years, including the islands of Samos and Chios (region of North Aegean) for the first time.

This 2-year study was carried out in a total of 12 regions of the country and 15 mosquito species of medical and veterinary importance and known vectors of several pathogens were recorded [22,23], including the major WNV vector *Cx. pipiens* s.l., the arboviral vector *Ae. albopictus* and the competent malaria vectors *An. maculipennis, An. sacharovi* [3,4,29,30].

*Cx. pipiens* s.l. was by far the most abundant species consistently present in all regions, with the highest mean densities being recorded in the region of Central Macedonia. These findings should be interpreted with caution, as the collected mosquito species composition depends also on the type or mosquito trap used and the traps’ location, i.e., their proximity to productive mosquito breeding sites; factors that can cause limitations in the direct comparison of mosquito abundance between regions. *Cx. pipiens* s.l. is considered the main vector of West Nile virus (WNV) in Europe [15,18] and Greece [19,31], and is associated with disease occurrence and burden in affected and vulnerable areas of the country. Seasonal outbreaks of West Nile virus disease reported in neighboring countries (Serbia, Italy, France, Romania, Hungary, Russia, and Spain) have also been associated with this species [21].

Τhe Asian tiger mosquito, *Ae. albopictus*, is recorded in Greece since 2003. Monitoring and surveillance of this species are continuously performed in the country thereafter and so far it has been found in 11 regions surveyed [32,33,34,35]. In the present study, it was recorded in 7 out of 8 regions surveyed in 2019 and in 10 out of 11 in 2020. In the 2019–2020 national program, *Ae. caspius* was found to be abundant mostly in the regions of Central Macedonia and Western Greece, as in previous entomological studies [31,34]; however, its role as a disease vector in Greece and as a WNV vector is to be further investigated.

Regarding populations of Anopheles species, *An. maculipennis, An. sacharovi, An. superpictus, An. hyrcanus, An. claviger,* and *An. algeriensis* have been recorded in Greece from previous studies [24,25,34]. It is known that several Anopheles species are important malaria vectors and, in this study, we recorded a relatively small density of *An. maculipennis, An. sacharovi, An. claviger, An. hyrcanus* and *An. algeriensis*, in different regions. As these species are considered competent or potential malaria vectors [35,36], further analysis is required to assess their role and contribution to malaria transmission in endemic countries.

Cases of WNV infection have been recorded in humans in Greece with seasonal outbreaks, mainly from the end of June–early July to early mid-October, on an almost annual basis. In 2018, the largest number of cases of WNV infection was recorded in Europe, with their total number exceeding the total number of all cases recorded in the previous 7 years, between 2010 and 2017 [8]. In Greece, 317 WNV infection cases were diagnosed in 2018, with 243 cases of neuroinvasive disease (WNND), representing a 23% increase compared with 2010, the previous most intense season. Two major epicenters were identified in 2018, in Attica and in Central Macedonia regions [9].

During the 2019 transmission period, two hundred and twenty-seven (n = 227) laboratory-diagnosed cases of WNV infection have been reported to the NPHO. The majority of human cases were recorded in the regions of East Macedonia and Thrace, Central Macedonia, Thessaly, and Attica. In 2020, the number of reported cases of WNV disease in Greece was one hundred and forty-five (n = 145), with all cases being recorded in the above-mentioned four regions of the country [10]. Further analysis of the WNV infectious indicators on a larger scale would be needed in order to identify possible associations with the human WNV infection burden at the local level.

## 5. Conclusions

This study highlights the importance of mosquito surveillance programs, mosquito species identification and abundance monitoring, and WNV circulation studies in Greece and the need for the establishment of a sustainable nationwide systematic vector surveillance network, as vectors related to human vector-borne disease cases, clusters or outbreaks are present in many areas in the country.

A human-animal-arthropod (One-Health) approach for WNV surveillance is of major importance for the early detection of human cases and the prompt implementation of targeted public health response measures including effective vector control strategies for WNV.

## Figures and Tables

**Figure 1 tropicalmed-08-00001-f001:**
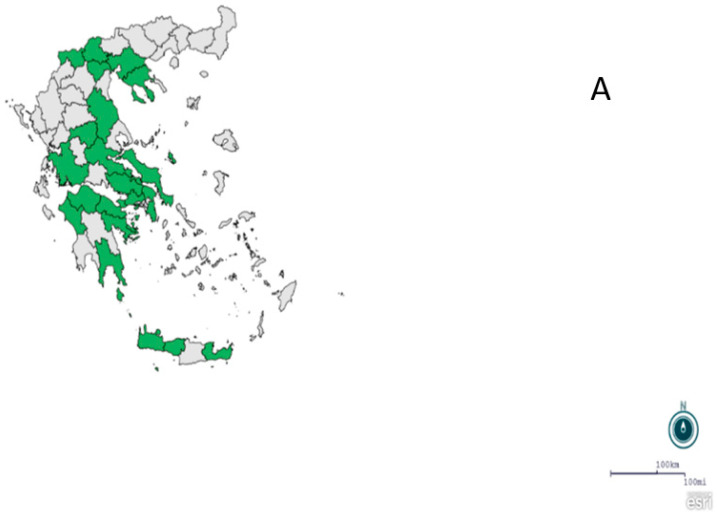
(**A**) Map showing the regional units where traps placed; (**B**) spot map showing the trapping points (n = 34), National Entomological Surveillance Program of NPHO, Greece, 2019.

**Figure 2 tropicalmed-08-00001-f002:**
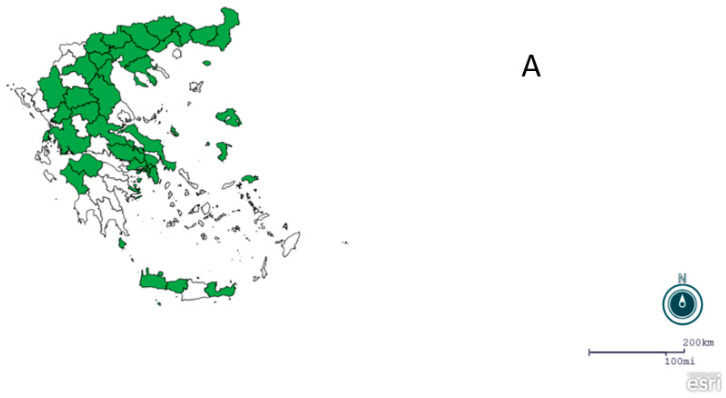
(**A**) Map showing the regional units where traps placed; (**B**) spot map showing the trapping points (n = 59), National Entomological Surveillance Program of NPHO, Greece, 2020.

**Figure 3 tropicalmed-08-00001-f003:**
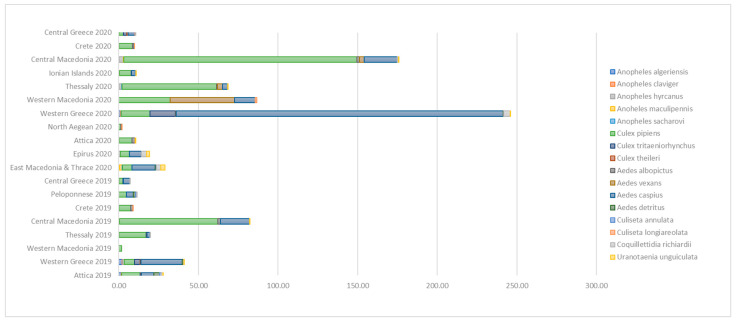
Mean number of collected mosquitoes per sampling, per species, by region and year, national entomological surveillance program of NPHO, Greece, 2019–2020.

**Table 1 tropicalmed-08-00001-t001:** Mean (SD; min–max) numbers of female *Cx. pipiens* s.l. mosquitoes per sampling, per year and region, Greece, 2019–2020.

Region	2019	2020	*p*-Value for the 2020 vs. 2019 Comparison
Central Greece	2.65 (6.56; 0–37) ^§^	2.61 (6.56; 0–40) ^§^	0.551
Peloponnese	4.42 (3.35; 0–13) ^†^	-	Not estimable
Crete	5.33 (7.58; 0–29) **	7.42 (9.78; 0–39) ^†^	0.265
Western Greece	4.75 (6.69; 0–33) **	16.52 (19.71; 0–82) ^†^	0.0001
Thessaly	15.33 (32.14; 0–160) *	58.17 (200.23; 0–1441) ^†^	0.002
Central Macedonia	60.19 (153.30; 0–948) ^†^	143.79 (269.19; 0–1283) ^†^	0.027
Western Macedonia	0.43 (0.51; 0–1)	30 (26.46; 0–95) ^†^	0.0002
Attica	9.67 (22.04; 0–158) ^†^	4.64 (6.20; 0–35) ^†^	0.0005
Epirus	-	1.75 (3.44; 0–11) ^§^	Not estimable
East Macedonia and Thrace	-	3.31 (4.31; 0–17) ***	Not estimable
Ionian Islands	-	3.6 (3.78; 0–8) *	Not estimable
North Aegean	-	0.77 (1.59; 0–7)	Not estimable

^§^ *p* ≥ 0.05; * *p* < 0.05; ** *p* < 0.01; *** *p* < 0.001; ^†^ *p* < 0.0001 versus the region with lowest mean number (Western Macedonia for 2019; North Aegean for 2020), *p*-values derived from Mann–Whitney–Wilcoxon test for independent samples.

**Table 2 tropicalmed-08-00001-t002:** Number of tested (Nt) and positive (Np) for West Nile virus (WNV) *Culex pipiens* s.l. mosquito pools, MIR and MLE (95%), by region and year, Greece, 2019–2020.

Regions	West Nile Virus						
	2019			2020			*p*-Value for the 2020 vs. 2019 Comparison in Np/Nt Rates ^†^
	Np/Nt	MIR (95% CI)	MLE (95% CI)	Np/Nt *	MIR (95% CI)	MLE (95% CI)	
Central Greece	1/12	8.47 (0.00–25.01)	9.17 (0.52–49.16)	0/19	0.00 (NE)	0.00 (0.00–25.98)	0.387
Peloponnese	1/16	9.71 (0.00–28.64)	9.89 (0.57–48.49)	−	-	-	Not estimable
Crete	2/14	14.18 (0.00–33.70)	13.89 (2.72–44.46)	0/30	0.00 (NE)	0.00 (0.00–12.60)	0.096
Western Greece	0/11	0.00 (NE)	0.00 (0.00–24.84)	0/40	0.00 (NE)	0.00 (0.00–4.95)	Not estimable due to zero events
Thessaly	1/12	2.81 (0.00–8.31)	2.61 (0.17–12.68)	4/75	1.04 (0.02–2.06)	1.07 (0.35–2.55)	0.533
Central Macedonia	11/59	2.31 (0.95–3.68)	2.58 (1.41–4.43)	8/171	0.46 (0.14–0.78)	0.47 (0.22–0.90)	0.002
Western Macedonia	1/4	250.0 (0.00–674.34)	250.0 (15.22–737.44)	0/10	0.00 (NE)	0.00 (0.00–9.14)	0.286
Attica	1/19	2.64 (0.00–7.80)	2.51 (0.16–12.11)	0/110	0.00 (NE)	0.00 (0.00–5.18)	0.147
Epirus	−	-	-	0/5	0.00 (NE)	0.00 (0.00–88.75)	Not estimable
East Macedonia and Thrace	−	-	-	0/26	0.00 (NE)	0.00 (0.00–21.68)	Not estimable
Ionian Islands	−	-	-	0/3	0.00 (NE)	0.00 (0.00–124.18)	Not estimable
North Aegean	−	-	-	0/23	0.00 (NE)	0.00 (0.00–56.28)	Not estimable
No. of WNV positive/tested mosquito pools	18/147			12/512			

* Np/Nt = number of WNV positive pools/numbers of tested pools.; NE: not estimable due to zero positivity; ^†^ *p*-values derived from Fisher’s exact test.

## Data Availability

The data that support the findings of this study are available from the corresponding author upon reasonable request.

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
