# Peer review of "Entomological Surveillance Activities in Regions in Greece: Data on Mosquito Species Abundance and West Nile Virus Detection in Culex pipiens Pools (2019–2020)"

_tropicalmed, 2022, doi:10.3390/tropicalmed8010001_

Round 1

Reviewer 1 Report

This is a nice and heavy work and the paper reports interesting results that deserve attention.

However, two main comments:

1. Figure 3: Needs to improve the lisibility

2. Results in Table 1 should be completed with MIR (mean infection rate) and TIR (true infection rate) analyses

The authors should address these comments before consideration of publication.

Author Response

Dear Reviewer,

thank you very much for your kind comments and your interest and attention in our work.

As requested, Figure 3 is now presented in another way to improve both lisibility and data presentation.

In the Results section,in Table we have incorporated columns with the MIR and MLE infection rates and analyses.

Reviewer 2 Report

Comments on the Manuscript tropicalmed-1978637 entitled " Entomological surveillance activities in region in Greece: Data on mosquito species abundance and West Nile virus detection 3 in Culex pipiens pools (2019-2020). "

The paper reports the 2019 and 2020 entomological data and WNV detected of a mosquito-based surveillance system in Greece. Because these data are important for better understanding of WNV epidemiology and the effective and efficient control of outbreaks caused by this virus in this area, this study is needed and of great importance. However, the paper needs a strong scientific and English editing. The following points need to be clarified before:

Major comments:

-       The sampling protocol was not well described. How many traps and sampling sites were used? For how many times, weekly or monthly? Please give us the sampling effort.

-       Describe the temporal and spatial dynamics of the main vector and the virus. Were the differences statistically significant

-       statistical analyses are missing. were the mean or median numbers and infection rates of Cx. pipiens females statistically different or not

Minor comments:

L60: give a ref.

L77: NPHO???

L91: the section beginning here belong to the previous paragraph

L98-99: why the authors did not separate members of the pipiens complex by molecular method?

L124: how mosquitoes already identified ad Cx. pipiens could be pooled by species and gender?

Figure 3: scientific names should be italicized. the name of mosquito genera should be fully written for the first appearance

Table 1: mosquitoes minimum infection rates should be calculated and compared spatially and temporally

L182-183 and 188: correct the typos

L200-206: this section has no relation with WNV and should be deleted.

Author Response

Dear Reviewer,

thank you very much for considering our work as important and for your useful comments. The manuscript has been edited accordingly to the best of our efforts.
Regarding your major comments:

-The part regarding the sampling protocol in now described in detail as far as the number of traps, sampling sites and frequency is concerned.

- We have performed statistical analysis in order to describe the temporal and spatial dynamics of the main vector, Culex pipiens  and West Nile virus. Methodology applied is described in the Ms & Ms section and the results are presented  in Results section.

- Statistical analyses regarding the mean numbers and infection rates of Culex pipiens females have been performed. Methodology applied is described in the M and M section and the results are presented  in Results section.

Regarding your minor comments,

They have all been addressed in the manuscript.

As far as your comment about performing molecular methodology in order to separate members of the pipiens complex, our main goal is monitoring and surveillance. Apart from the high cost, it would be laborious and time consuming and we would not benefit of the real time reporting both of mosquitoes' abundance and virus presence in the pools.

Reviewer 3 Report

1. Authors have used assays to detect WNV lineage and NS2A. However results have been discussed based on just WNV positive pools. Results should be based on these two assays and discussion should also be based on the two assays.

2.Why other potential vectors, e.g. Ae. albopictus, of WNV collected have not been screened for the presence of  WNV

Author Response

Dear Reviewer,

thank you for considering our manuscript and for your useful comments.

We have revised it accordingly.

Regarding your comment on the two real time PCR assays, one for routine detection and a second only for further verification of positive samples, it is now described in more detail and we hope that it is clarified.

In Greece, it is known that the main vector of WNV is Cx. pipiens. In the past our group (Refs 20 and 30) and others (Mavridis et al. Detection of West Nile Virus - Lineage 2 in Culex pipiens mosquitoes, associated with disease outbreak in Greece, 2017. Acta Trop. 2018 Jun;182:64-68; Bisia et al.The abundance and diversity of West Nile virus mosquito vectors in two Regional Units of Greece during the onset of the 2018 transmission season, have also examined for the presence of WNV Ae. albopictus, Aedes caspius and other species collected, with both positive and negative findings, however they have never been considered as competent vectors. Therefore, for the scope of the national entomological surveillance program and in order to be cost effective and prompt, we have focused our screening on the established as the main vector of the virus in the country.

Round 2

Reviewer 1 Report

The authors have satisfactorily addressed my queries.